# Medium-Chain Fatty Acids Extracted from Black Soldier Fly (*Hermetia illucens*) Larvae Prevents High-Fat Diet-Induced Obesity In Vivo in C57BL/6J Mice

**DOI:** 10.3390/ani15101384

**Published:** 2025-05-10

**Authors:** Kyu-Shik Lee, Min-Gu Lee, Kyuho Jeong, Eun-Young Yun, Tae-Won Goo

**Affiliations:** 1Department of Pharmacology, College of Medicine, Dongguk University, Gyeongju 38766, Republic of Korea; there1@dongguk.ac.kr (K.-S.L.); lmg936122@gmail.com (M.-G.L.); 2Department of Biochemistry, College of Medicine, Dongguk University, Gyeongju 38766, Republic of Korea; khjeong@dongguk.ac.kr; 3Department of Integrative Bio-Industrial Engineering, Sejong University, Seoul 05006, Republic of Korea

**Keywords:** black soldier fly larvae, medium-chain triglycerides, obesity prevention, high-fat diet

## Abstract

This investigation aimed to analyze the effect of medium-chain fatty acids (MCFAs) derived from black soldier fly larvae (BSFL-MCFAs) in male C57BL/6J mice fed a high-fat diet (HFD). The results suggest that BSFL-MCFAs may exert anti-obesity activity and contribute to improvements in blood lipid parameters in HFD-induced obese mal C57BL/6J mice. These findings demonstrate the potential of BSFL-MCFAs as feed and food ingredients for the prevention of obesity and related metabolic disorders in both animals and humans. However, further studies are needed to confirm these effects and to elucidate the underlying mechanisms.

## 1. Introduction

Obesity is a chronic disease caused by excessive fat accumulation resulting from an imbalance between energy intake and expenditure [1]. The global prevalence of obesity has risen sharply over the past few decades [2]. According to a meta-analysis performed by the Non-Communicable Disease Risk Factor Collaboration, obesity in adults has more than doubled globally, from 13.8% in women and 10.7% in men in 1990 to 28.3% in women and 26.9% in men by 2022, respectively [3]. Obesity is a risk factor for various disorders, including certain cancers, metabolic disorders such as diabetes and hyperlipidemia, cardiovascular diseases (e.g., hypertension and arteriosclerosis), and cerebrovascular events [4,5,6,7]. Furthermore, obesity contributes to musculoskeletal disorders and a reduced quality of life [8]. Also, obesity-related complications pose a risk to animal health [9]. Therefore, the prevention of obesity is important for maintaining health and improving the quality of life for humans and animals.

While obesity can be treated using various approaches, depending on its causes, severity, and associated comorbidities, lifestyle modifications—particularly dietary control and increased physical activity—remain the primary strategy for its prevention and management [10,11]. However, extreme caloric restriction, often used in dietary interventions, may lead to adverse effects, such as a decrease in the basal metabolic rate, bone loss, hormone imbalance, nutrient deficiencies, and hypoglycemia [12,13,14,15]. Therefore, recent efforts have focused on dietary composition rather than reducing total caloric intake, aiming to enhance energy metabolism and reduce fat accumulation without compromising overall health [16,17].

Medium-chain fatty acids (MCFAs) have attracted considerable attention in this context, MCFAs, which are triglycerides found in various plant oils such as olive, palm, and coconut oils, are a fat source better absorbed and metabolized by long-chain triglycerides [18,19,20,21]. Numerous preclinical and clinical studies have shown that MCFAs enhance hepatic thermogenesis in mice [22], increase energy expenditure [23], and improve metabolic profiles by reducing adiposity and insulin resistance [23,24,25,26,27]. Despite these benefits, the supply of traditional plant-based oil has been increasingly affected by climate change-induced agricultural instability, leading to supply shortages and price volatility [28,29,30,31,32].

In response to these challenges, research is being conducted to explore the use of insect oils as alternatives [33,34]. The larvae of the black soldier fly (*Hermetia illucens*) (BSFL) are known as waste-reducing insects [35]. BSFL are primarily used as animal feed, but recent studies have suggested their potential as a novel nutritional source with reports highlighting antimicrobial activity [34,36,37,38]. Furthermore, studies have shown that obese Zucker rats fed BSFL have reduced hepatic lipid synthesis and fatty acid production and that the lipids in BSFL contribute to lowering the liver fat concentration [34]. It would strengthen the nutritional and industrial relevance of BSFL lipids to compare them directly with conventional MCT sources such as coconut and palm oils. Coconut oil typically contains around 45–53% lauric acid (C12:0), while palm kernel oil contains about 48% [39,40]. In contrast, BSFL lipid profiles are highly modifiable through dietary manipulation, with studies showing that lauric acid content in BSFL oil can be increased to over 70% [41]. Previous studies have demonstrated that BSFL-derived oils, which are rich in MCFA such as lauric acid, can influence metabolic health parameters in animals. For example, BSFL oil supplementation in broiler diets has shown to improve gut microbiota and lipid metabolism [42], while studies in pigs and rodents have suggested potential effects on immune modulation and hepatic lipid reduction [43,44]. This offers a distinct advantage in tailoring fatty acid composition based on application needs. Moreover, BSFL oil production exhibits a significantly lower unit cost compared to plant-based MCT sources, due to shorter production cycles, the use of low-value organic waste as feed, and lower land and water use requirements. However, to date, few investigations have explored the metabolic effects of specific lipid components—particularly MCFAs—derived from BSFL. Therefore, based on these findings, this investigation aimed to quantify the MCFA levels in BSFL-derived lipids and to evaluate the potential of MCFAs extracted from BSFL (BSFL-MCFAs) in preventing high-fat diet-induced obesity in male C57BL/6J mice.

## 2. Materials and Methods

### 2.1. Rearing BSFL

The BSFL were reared at the Greenteko Corporation (Hwaseong, Republic of Korea). Briefly, BSFL were reared for 21 days under controlled conditions (26 ± 1 °C and 50% relative humidity). A food waste and waste cooking oil mixture was fermented for 5 days using effective microorganisms, following the method reported by Lee et al., and was used as BSFL feed [45]. The term “effective microorganisms” refers to a mixed culture of beneficial microbes, consisting primarily of lactic acid bacteria, yeasts, and photosynthetic bacteria.

### 2.2. Extraction of Crude Oil from BSFL and MCFA Preparation

The extraction of crude oil from BSFL and preparation of MCFAs from the crude oil were performed by Greenteko Corporation. The harvested BSFL were dried in a 700 W microwave for 8.5 min. To extract the crude oil, the dried BSFL were pressed at 120 °C (20 MPa), and the liquid extract was collected as crude oil [45]. Subsequently, the crude oil was mixed with water and sulfuric acid in a ratio of 7.0:1.0:0.1 (water/crude; oil/sulfuric acid), and the mixture was reacted at 160 °C for 20 h to prepare the fatty acids. The fatty acid composition was analyzed using gas chromatography (GC)/mass spectrometry at the Korea Quality Testing Institute (Suwon, Republic of Korea). The analysis was performed using an Agilent 7890B GC system equipped with a DB-23 capillary column (60 m × 0.25 mm i.d., 0.25 μm film thickness; Agilent Technologies, Santa Clara, CA, USA). The oven temperature was programmed to start at 130 °C (held for 1 min), increased to 170 °C at 6.5 °C/min, then to 215 °C at 2.75 °C/min, and finally to 230 °C at 40 °C/min (held for 5 min). Helium was used as the carrier gas at a constant flow rate of 1.0 mL/min. Fatty acid methyl esters were prepared according to the standard AOCS Ce 2-66 method [46]. MCFAs were isolated from the prepared fatty acids by vacuum distillation (<7.5 mmHg). Briefly, the fatty acids were heated at 100 °C under vacuum to remove moisture. The temperature was then gradually increased from 200 °C to 245 °C under vacuum, for the evaporation and separation of MCFAs [47].

### 2.3. Animal Experiments

All animal experiments were approved by the Animal Use and Care Committee of Dongguk University (IACUC-2022-09, Gyeongju, Republic of Korea). Forty-five 7-week-old male C57BL/6J mice were purchased from Hana Biotech (Pyeungtaek, Republic of Korea) and stabilized for one week on a normal diet (ND). Mice were then randomly divided into the ND, high-fat diet (HFD), and HFD-containing BSFL-MCFAs (HFD_MCFA) groups (15 mice/group). Each group was further subdivided into three subgroups of five mice each. The mice received ND, HFD, or HFD_MCFA ad libitum for 80 days. For each subgroup, the total body weight and feed intake were measured every 2–3 days during the experiment, and the average body weight and feed intake per mouse were calculated. The mean body weights and feed intake per mouse from each subgroup were aggregated and averaged for each group, and the standard deviations of the average body weight and feed intake per mouse were calculated. The food efficiency ratio (FER) was determined based on the initial and final average body weights, as well as the daily feed intake. During the experiment, mice were housed in polysulfone cages with wood chip bedding and provided with environmental enrichment (e.g., paper rolls and plastic huts) to promote exploratory behavior and reduce stress. Housing conditions were maintained at 22 ± 1 °C, 50 ± 5% relative humidity and a 12 h light/dark cycle. After 80 days, the mice were sacrificed by inhalation of Zoletil50 (Virbac Korea, Seoul, Republic of Korea) after fasting for 12 h. Blood and hypothalami were collected to analyze the biological composition and gene expression of endoplasmic reticulum (ER) stress markers.

### 2.4. Animal Diet

ND (product #: D12450B) and HFD (product #: D12492) were purchased from Research Diets (New Brunswick, NJ, USA), and HFD_MCFA was prepared by DOOYEOLBIOTECH (Seoul, Republic of Korea). The compositions of the HFD and HFD_MCFA diets are presented in Table 1. All diets were designed to be isocaloric (~4000 kcl/kg) and matched for macronutrient distribution. HFD and HFD_MCT contained ~60% kcal from fat, 20% from protein, and 20% from carbohydrates, with lard replaced by MCFAs in HFD_MCFA. ND provided ~10% kcal from fat, 20% from protein, and 70% from carbohydrates. A summary of the proximate macronutrient composition is shown in Appendix A.

### 2.5. Measurement of Blood Lipid Levels

Triglyceride (TG), total cholesterol (TC), and high-density lipoprotein cholesterol (HDL) levels in the blood were analyzed by the Korea Non-Clinical Technology Solution Center (Seongnam, Republic of Korea) using standard enzymatic colorimetric analysis based on the Trinder reaction [48,49]. Low-density lipoprotein cholesterol (LDL) levels were determined using the Friedewald formula. The formula is as follows: [LDL (mg/dL) = TC (mg/dL) − HDL (mg/dL) − TG (mg/dL)]/5 which is valid when TG < 400 mg/dL [50]. Based on the lipid levels, the atherogenic index (AI) and cardiac risk factor (CRF) were calculated using the Lauer and Rosenfeld formulae, respectively. The formulae are as follows: AI = ([TC] − [HDL-c])/[HDL-c] and CRF = [TC]/[HDL-c] [51].

### 2.6. Analysis of Liver Injury Indicators in the Blood

We evaluated the activities of aspartic acid transaminase (AST) and alanine transaminase (ALT) in the blood as liver damage indicators. The activities were assessed by the Korea Non-Clinical Technology Solution Center using the Reitman–Frankel method.

### 2.7. Analysis of Kidney Function Indicators in the Blood

Blood urea nitrogen (BUN) and creatinine levels were measured as indicators of kidney function. BUN and creatinine levels were assessed by the Korea Non-Clinical Technology Solution Center using the enzymatic methods of Talke and Schubert and Jeff reaction, respectively [52,53]. BUN was quantified by the enzymatic conversion of urea to ammonia using urease, followed by a colorimetric reaction. Creatinine was measured via the Jaffe reaction, in which creatinine reacts with alkaline picrate to form a colored complex, which is quantified spectrophotometrically.

### 2.8. Evaluation of Glucose and Leptin Concentrations in the Blood

We analyzed the blood glucose and leptin levels, because circulating glucose and leptin concentrations are closely associated with obesity. The analyses were performed using a Glucose Colorimetric Detection Kit and mouse leptin ELISA kit (Invitrogen, Thermo Fisher Scientific; Waltham, MA, USA), respectively.

### 2.9. Quantitative Real-Time Polymerase Chain Reaction (PCR)

We analyzed the expression of obesity-associated ER stress markers (Grp7, Chop, Xbp-1s, Atf4, and ErdJ4) in the hypothalamus. The mouse hypothalamus was homogenized, and the total RNA was extracted using TRIzol reagent (Invitrogen, Thermo Fischer Scientific, Waltham, MA, USA), according to the manufacturer’s instructions. Single-strand cDNA was then synthesized using a high-capacity reverse transcription kit (Applied Biosystems, Thermo Fisher Scientific, Waltham, MA, USA), and the relative gene expression of ER stress markers was determined using the cDNA and real-time PCR. GAPDH was used as an internal control. The primer sequences for ER stress markers and GAPDH are presented in Table 2. The primers used in this investigation were identical in sequence to those previously reported [54].

### 2.10. Statistical Analysis

Statistical analysis of the data was performed using one-way analysis of variance (ANOVA), followed by Tukey’s honestly significant difference test. SPSS software (version 20.0; SPSS, Inc., Chicago, IL, USA) was used for the analysis, and *p*-values < 0.05 were considered significant.

## 3. Results

### 3.1. Fatty Acid Compositon of BSFL Crude Lipids

To prepare the BSFL-MCFAs, the fatty acid composition of the crude lipids extracted from BSFL was analyzed. Lauric acid was the most prevalent among the crude lipids (51.1%), followed by stearic, palmitic, and oleic acids (12.8%, 12.1%, and 11.1%, respectively) (Table 3). Additionally, fatty acids with a carbon chain length of ≤12 comprised 53.2% of the crude lipids. Furthermore, 85.6% of crude lipids were saturated fatty acids. Consequently, these findings suggest that the crude lipids extracted from BSFL contain >50% MCFAs, implying an association between high MCFA levels in BSFL crude lipids and their weight reduction effects.

### 3.2. Effect of MCFAs on Weight Gain of Mice Fed HFD

We evaluated the effects of BSFL-MCFAs on weight gain in mice. Mice in the HFD group showed the greatest increase in body weight, whereas the weight gain was significantly attenuated in the HFD_MCFA group (Figure 1). Notably, the weight gain in mice in the HFD_MCFA group was lower than that in the ND group. The highest daily food intake was observed in the ND group, with no significant difference in food intake between the HFD and HFD_MCFA groups (Figure 1b). On the basis of these findings, we calculated the FER for each group. The FER in the HFD_MCFA group was lower than that in the ND and HFD groups (Figure 1b). These results demonstrate that BSFL-MCFAs may help suppress weight gain in HFD-fed mice.

### 3.3. Effect of MCFAs on Blood Lipid Parameters

To evaluate the effects of MCFAs on blood lipid parameters, the TG, TC, LDL, and HDL concentrations were measured. The TG, TC, and LDL levels were significantly elevated in the HFD group; however, the increases were attenuated in the HFD_MCFA group. In contrast, the HDL levels in both the HFD and HFD_MCFA groups were higher than that in the ND group (Figure 2a). Based on these results, we calculated the AI and CRF. The AI and CRF were significantly increased in the HDF group, whereas these increases were suppressed in the HFD_MCFA group (Figure 2b). Moreover, the AI and CRF in the HFD_MCFA group were lower than those in the ND group. These findings suggest that BSFL-MCFAs may help prevent atherogenesis and cardiac disease development.

### 3.4. Effect of MCFAs on Liver Injury and Kidney Function

An HFD can induce metabolic changes that lead to liver and kidney injury, as evidenced by alterations in serum biomarkers such as AST, ALT, BUN, and creatinine [55,56,57,58,59,60]. Therefore, we analyzed these markers to investigate the effects of MCFAs on the liver and kidney functions in HFD-fed mice. The results showed no significant increase in AST, ALT, BUN, or creatinine levels in the HFD group compared to those in the ND group (Figure 3). Interestingly, ALT levels in the HFD_MCFA group were significantly lower than those in the ND and HFD groups, suggesting a potential protective effect of MCFAs on liver function (Figure 3a). Additionally, although the HFD_MCFA group exhibited a significant increase in BUN levels compared with the HFD group, no significant difference was observed between the ND and HFD_MCFA groups (Figure 3b). These findings demonstrate that BSFL-MCFAs may exert differential effects on serum biomarkers associated with liver and kidney functions in HFD-fed mice, particularly by reducing ALT levels.

### 3.5. Effect of MCFAs on Blood Glucose and Leptin Levels

Animal experiments have elucidated the relationship between an HFD and increased serum glucose, and the relationship between obesity and increased serum glucose levels has been reported [61,62,63,64,65]. Therefore, we measured blood glucose and leptin levels to examine the effect of MCFAs on obesity. Significant increases in the blood glucose and leptin levels were observed in the HFD group. In contrast, the blood glucose and leptin levels in the HFD_MCFA group were significantly lower than those in the HFD group (Figure 4). Furthermore, lower leptin levels were observed in the HFD_MCFA group than in the ND group. These results suggest that BSFL-MCFAs may help prevent HFD-induced obesity.

### 3.6. Effect of MCFAs on the Expression of ER Stress Markers

Several studies have reported that hypothalamic ER stress is closely associated with obesity and leptin resistance [66,67,68,69,70,71,72]. Therefore, we investigated changes in the expression of ER stress markers, such as Grp78, Erdj4, Xbp-1s, Atf4, and Chop, in the hypothalamus. The mRNA levels of ER stress markers were significantly increased in the HFD group, implying that the increase in body weight in HFD-fed mice was associated with obesity (Figure 5). In contrast, we observed decreased mRNA levels of ER stress markers in the HFD_MCFA group, suggesting the prevention of obesity by MCFAs (Figure 5). These findings indicate that BSFL-MCFAs may exhibit anti-obesity effects in HFD-fed C57BL/6J mice.

## 4. Discussion

BSFL have been utilized in agricultural and animal feed applications due to their high protein and lipid levels of approximately 42% and 35%, respectively [73]. Many studies have shown that lauric acid is the major fatty acid component in BSFL lipids [74,75,76,77,78]. We also found that the most prevalent fatty acid component of BSFL lipids was lauric acid [45]. Many studies have demonstrated the beneficial effects of lauric acid and triglycerides on metabolic disorders, including obesity, fatty liver, and diabetes [79,80,81,82,83]. In this investigation, mice supplemented with BSFL_MCFAs exhibited significantly reduced weight gain (Cohen’s *d* value: 11.68, effect-size *r*: 0.99) and FER (Cohen’s *d* value: 6.89, effect-size *r*: 0.96), without no significant change in food intake (Figure 1). Notably, both the weight gain and FER in the HFD_MCFA group were also lower than those observed in the ND group (Figure 1b). These findings are consistent with previous reports indicating that MCFA intake may increase thermogenesis and enhance energy expenditure, thereby contributing to the reduction in body weight in both animals and humans [22,23,84,85]. Taken together, these results suggest that the anti-obesity potential of BSFL-MCFAs may be associated with thermogenic activation and metabolic rate enhancement.

In this investigation, only male C57BL/6J mice were used to minimize potential variability associated with hormonal fluctuations in females, particularly those related to the estrous cycle, which can influence metabolic outcomes. This sex-specific selection is consistent with the design of many preclinical metabolic studies that aim to establish proof-of-concept efficacy. However, we acknowledge this as a limitation of the current study, and we have emphasized the importance of including female animals in future experiments to improve the generalizability and translational relevance of the findings.

Obesity is known to be associated with increased TC, TG, and LDL levels in the blood, which are established risk factors for metabolic disorders and cardiovascular diseases [86]. Accordingly, preventing obesity is a well-established strategy to improve lipid profiles and to reduce cardiovascular risk. In this study, significantly lower TC (Cohen’s *d* value: 2.97, effect-size *r*: 0.83), TG (Cohen’s *d* value: 1.26, effect-size *r*: 0.53), and LDL (Cohen’s *d* value: 1.30, effect-size *r*: 0.0.55) levels were observed in the HFD_MCFA group than in the HFD group (Figure 2). Additionally, the AI and CRF in the HFD_MCFA group were also lower than those in both the ND and HFD groups. Interestingly, HDL levels were elevated in both the HFD and HFD_MCFAs groups. This may reflect a compensatory response to high-fat intake, as increased HDL is often observed in rodent models fed high-fat diets [87]. In particular, MCFAs have been reported to promote HDL biosynthesis and improve reverse cholesterol transport by enhancing hepatic expression of apolipoproteins and lipid transporters [88,89]. These observations are consistent with previous reports indicating the lipid-lowering effects of MCFAs in animal models [90,91,92], and are further supported by recent meta-analysis report showing that MCFA-enriched diets contribute to reduced body weight, improved lipid profiles in blood, and decreased insulin resistance in overweight or obese individuals [93]. Therefore, these results imply that the significantly lower levels of TC, TG, and LDL in the HFD_MCFA group may be closely associated with the attenuated weight gain caused by the inhibition of body fat accumulation.

Although several investigations have reported increased blood levels of liver and kidney injury markers following HFD consumption, such increases were not observed in this investigation (Figure 3). Previous investigations have shown increases in liver and kidney injury markers such as ALT, AST, BUN, and creatinine after ≥14 weeks of HFD administration in C57B/6J mice [59,94,95]. In contrast, the present study was limited to 80 days (approximately 12 weeks), which may not have been sufficient to induce the damage markers. Therefore, the unobserved increase in these markers in our investigation may be attributed to the relatively short duration of HDF administration.

Interestingly, the HFD_MCFA group exhibited the lowest ALT levels and highest BUN levels, whereas no significant changes were observed in AST and creatinine levels. Reductions in ALT levels may be partially influenced by decreases in muscle mass, and increases in BUN levels are often associated with enhanced protein catabolism. As shown in Figure 1, the HFD_MCFA group exhibited less weight gain than the ND group, although no actual weight loss was observed. Collectively, these findings suggest that the changes in ALT and BUN levels in the HFD_MCFA group are unlikely to be the result of liver or kidney damage. Instead, the reduced weight gain in the HFD_MCFA group likely results from the inhibition of fat accumulation rather than from a loss of muscle mass. Considering that increased BUN levels are a common outcome of high-protein diets, it is plausible that MCFAs enhance the metabolic breakdown of dietary proteins, thereby contributing to the observed increase in BUN levels.

In this study, we found that supplementation with BSFL-MCFAs was associated with improvement in several obesity-related metabolic parameters in mice fed HFD. Lauric acid (C12;0), the predominant MCFA in BSFL crud lipids, may have contributed to these effects through its known metabolic properties. The HFD_MCFA group showed reduced serum glucose (Cohen’s *d* value: 1.28, effect-size *r*: 0.54) and leptin (Cohen’s *d* value: 112.89, effect-size *r*: 0.99) levels compared to the HFD group, which may reflect improvements in glucose regulation and metabolic hormone balance (Figure 4). In addition, gene expression analysis revealed that BSFL_MCFAs intake was accompanied by decreased expression of several ER stress markers, including Grp78, Erdj4, Xbp-1s, Atf4, and Chop, suggesting a potential reduction in HFD-induced ER stress (Figure 5). MCFAs, including lauric acid, have been reported to attenuate ER stress via modulation of the PERK-eIF2α-ATF4 pathway [96,97]. ER stress has been implicated in metabolic dysfunctions associated with obesity, including insulin resistance, leptin resistance, and chronic inflammation in adipose and hepatic tissues [66,69,98,99,100,101,102]. Specifically, ER stress can impair leptin signaling by activating the SOCS3 pathway and suppressing JAK2-STAT3 signaling, leading to leptin resistance [103,104]. The lower expression of Chop and Atf4, in particular, may demonstrate a downregulation of stress pathways related to the PERK-eIF2α axis, which is known to mediate cellular stress and inflammatory signaling [105,106]. Interestingly, the leptin levels in the HFD-MCFA group were not only lower than those in the HFD group but also the significantly reduced compared to the ND group. While this finding may suggest that BSFL-MCFAs enhance leptin sensitivity or energy expenditure, further investigation is needed to clarify the underlying mechanisms, in potential effects on thermogenesis or metabolic rates. Furthermore, the reduced weight gain observed in the HFD_MCFA group likely reflects fat mass loss rather than muscle catabolism, as indicators of muscle damage (AST) and renal function (creatinine) remained stable across groups (Figure 3). Overall, the results suggest that BSFL-MCFAs supplementation may contribute to metabolic improvements in the context of diet-induced obesity, possibly through modulating ER stress and related endocrine responses. However, further studies are necessary to reveal these findings and to explore the mechanistic pathways involved, particularly those linking ER stress with systemic metabolic regulation (Figure 6).

## 5. Conclusions

In conclusion, the findings of this study suggest that BSFL-MCFAs may contribute to the prevention of obesity by inhibiting fat accumulation and may serve as an effective nutritional supplement that promotes weight management by increasing the basal metabolic rate. These results demonstrate that BSFL-MCFAs should be considered as a feed additive for mitigating obesity risk in companion animals, with potential applicability as a functional dietary ingredient.

## Figures and Tables

**Figure 1 animals-15-01384-f001:**
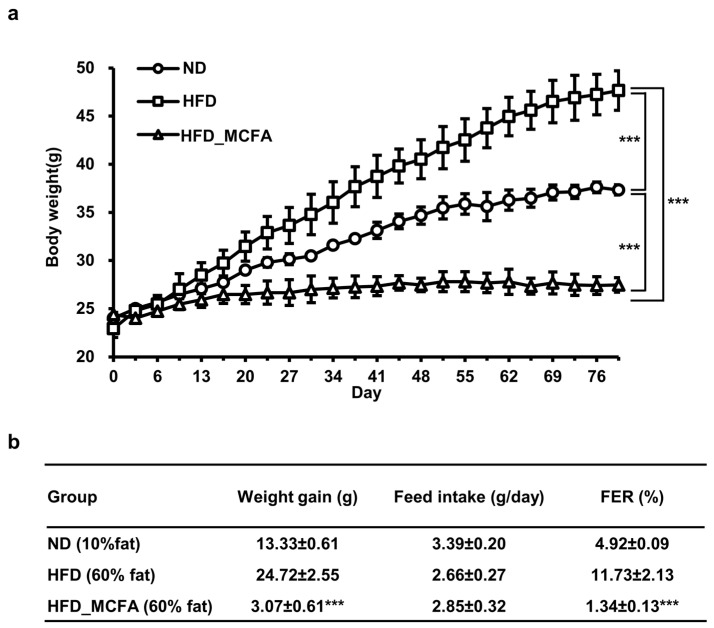
MCFAs attenuate the HFD-induced increase in body weight in mice with a decrease in FER. (**a**) The total body weight of each subgroup was measured every 2–3 days. Subsequently, the mean body weight per mouse was calculated for each subgroup. The subgroup averages were summed, and an overall mean was calculated to determine the average body weight per mouse of each group. Statistical differences between groups were determined through pairwise comparisons. *** *p* < 0.001. (**b**) Initial and final body weights of each subgroup were measured, and the mean initial and final body weights per mouse was calculated for each subgroup. The average initial body weight was subtracted from the average final body weight to determine the weight gain per mouse in each subgroup. The subgroup weight gains were averaged to calculate the overall weight gain per mouse for each group. The total feed intake for each subgroup was measured every 2–3 days, and the mean feed intake per mouse was calculated for each subgroup. The subgroup averages were then summed to calculate the overall feed intake per mouse for each group. The FER was calculated as a percentage by dividing the daily weight gain by the daily feed intake. *** indicates that there is a significant difference between the HFD_MCFA group and the HFD group at *p* < 0.001.

**Figure 2 animals-15-01384-f002:**
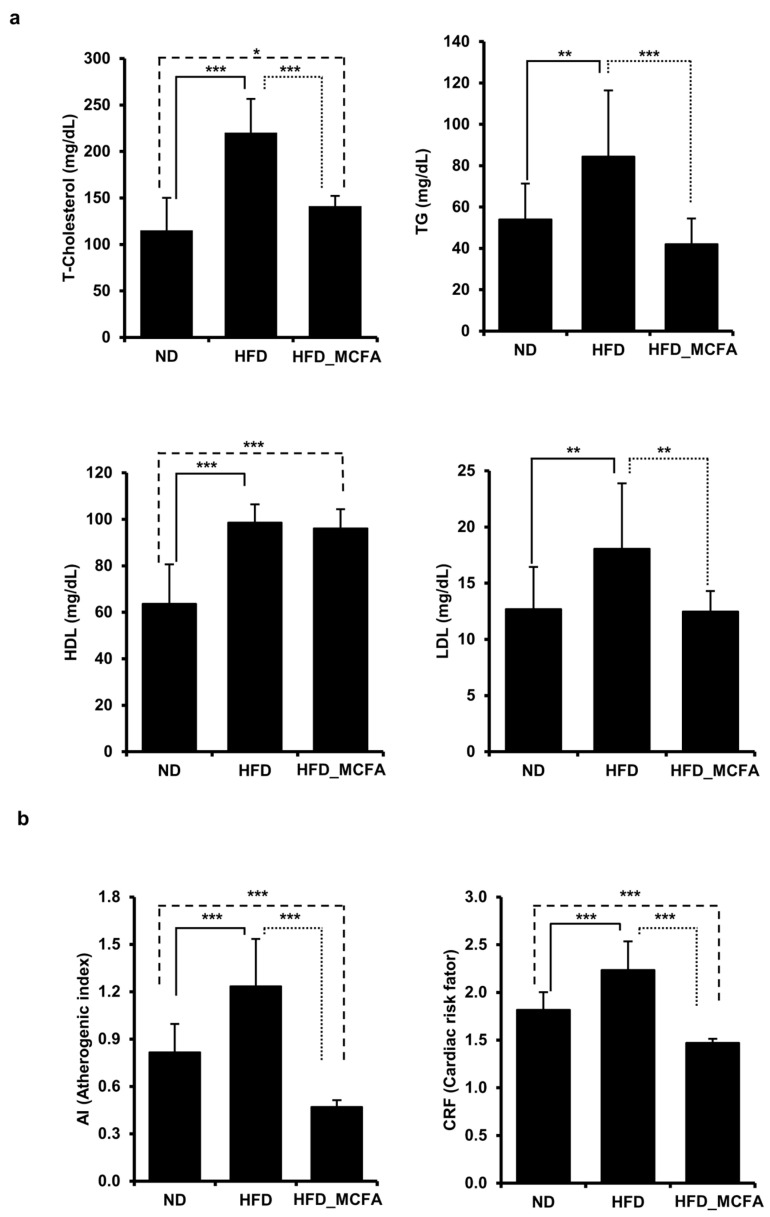
MCFAs prevent the HFD-induced increase in TC, TG, and LDL levels in mice, with decreases in the AI and CRF. (**a**) Blood samples were collected from each mouse after anesthesia, followed by euthanasia. TC, TG, HDL, and LDL levels of individual mice were measured and summed, and group averages were determined. * *p* < 0.05, ** *p* < 0.01, and *** *p* < 0.001. (**b**) The average AI and CRF for each group are presented. The AI and CRF were calculated based on the lipid levels of individual mice, and the group averages were then determined.

**Figure 3 animals-15-01384-f003:**
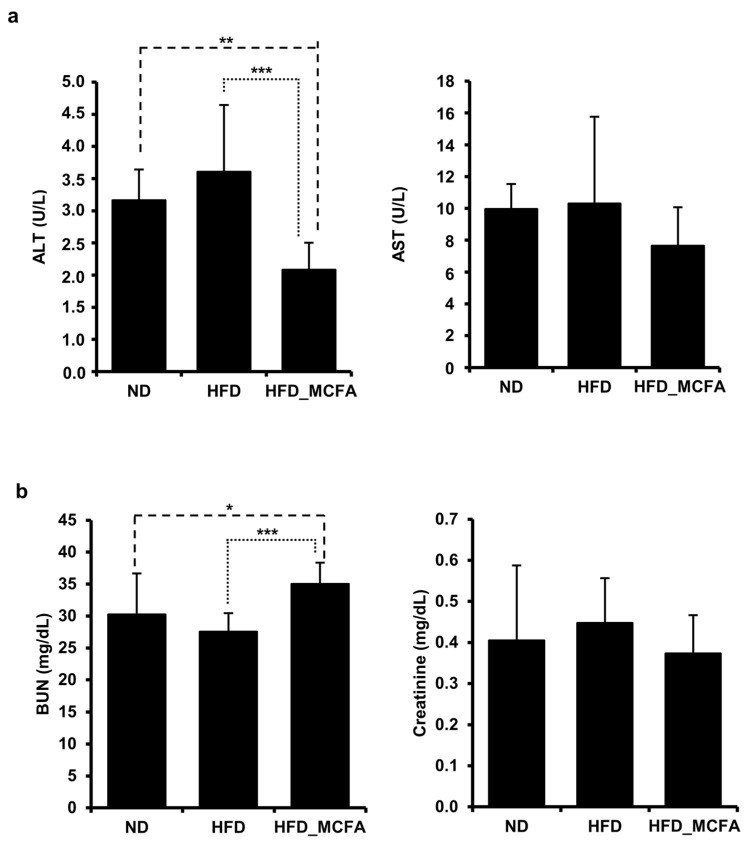
MCFAs decrease levels of the liver injury marker ALT. (**a**,**b**) Blood samples were collected from each mouse after anesthesia, followed by euthanasia. (**a**) The average ALT and AST levels for each group are presented. The ALT and AST levels of individual mice were measured and summed, and group averages were determined. ** *p* < 0.01 and *** *p* < 0.001. (**b**) Average BUN and creatinine levels for each group are presented. BUN and creatinine levels of individual mice were measured and summed, and group averages were calculated. * *p* < 0.05 and *** *p* < 0.001.

**Figure 4 animals-15-01384-f004:**
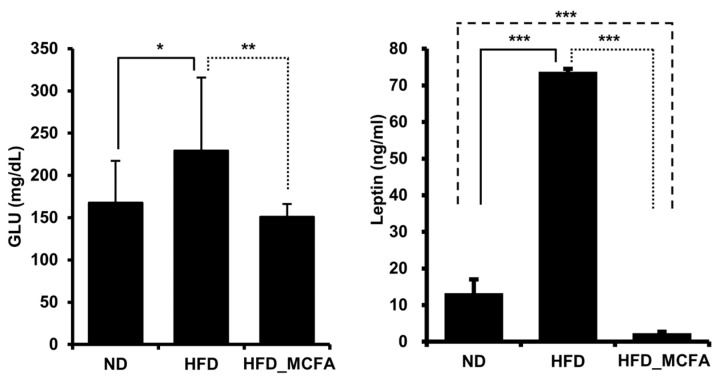
MCFAs significantly suppress the increases in blood glucose and leptin levels. Blood samples were collected from each mouse after anesthesia, followed by euthanasia. The average ALT and AST levels for each group are presented. ALT and AST levels of individual mice were measured and summed, and group averages were determined. * *p* < 0.05, ** *p* < 0.01, and *** *p* < 0.001.

**Figure 5 animals-15-01384-f005:**
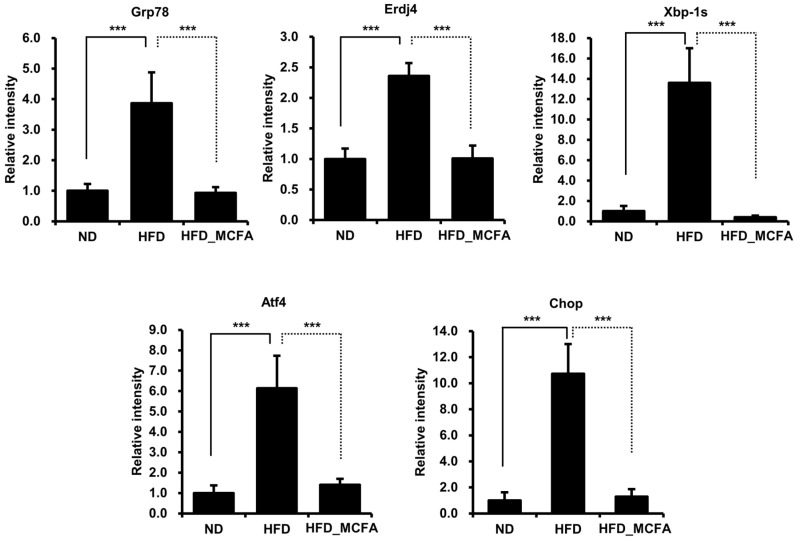
MCFAs significantly suppress the increases in the expression of ER stress markers. The mice were sacrificed under anesthesia, and the hypothalami were collected. Total RNA was extracted from the hypothalami, and equal amounts of total RNA from individual samples were pooled for each group. Gene expression levels of ER stress markers were then analyzed. *** *p* < 0.001.

**Figure 6 animals-15-01384-f006:**
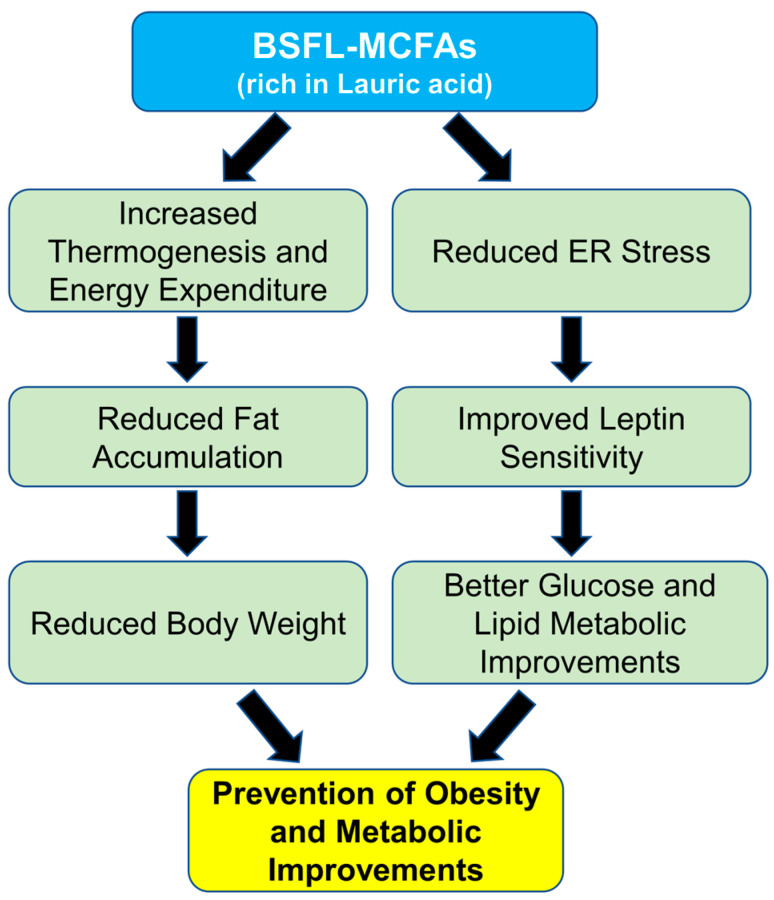
Schematic representation of the proposed mechanisms underlying the effects of MCFAs on HFD-induced obesity.

**Table 1 animals-15-01384-t001:** HFD and HFD_MCFA formulas.

Ingredients	Diet					
	ND	Calorie (kcal)	HFD (g)	Calorie (kcal)	HFD_MCFA (g)	Calorie (kcal)
Casein, 30 Mesh	200	800	200	800	200	800
L-Cystine	3	12	3	12	3	12
Maltodextrin 10	35	140	125	500	125	500
Sucrose	350	1379	68.8	275	68.8	275
Starch	315	1200	-	-	-	-
Cellulose, BW 200	50	0	50	0	50	0
Soybean oil	25	225	25	225	25	225
Lard	20	180	245	2205	-	2205
BSF-MCFAs	-	-	-	-	245	2205
Mineral Mix (S10026B)	50	0	10	0	10	0
Dicalcium phosphate	13	0	13	0	13	0
Calcium carbonate	5.5	0	5.5	0	5.5	0
Potassium citrate, 1H_2_O	16.5	0	16.5	0	16.5	0
Vitamin Mix (V10001C)	10	40	10	40	10	40
Choline bitartrate	2	0	2	0	2	0
FD&C Blue Dye #1	-	-	0.05	0	0.05	0
Dye, Yellow FD&C#5	0.05	0	-	-	-	-
Total	1095.05	3976	773.85	4057	773.85	4057

**Table 2 animals-15-01384-t002:** Primer sequences for the analysis of obesity-associated ER stress markers.

Name	Sequences/Tm		Amplicon Size (bp)
Chop	Forward	5′-CCACCACACCTGAAAGCAGAA-3′/61 °C	67
	Reverse	5′-AGGTGAAAGGCAGGGACTCA-3′/61 °C	
Grp78	Forward	5′-GGCCTGCTCCGAGTCTGCTTC-3′/65 °C	243
	Reverse	5′-CCGTGCCCACATCCTCCTTCT-3′/64 °C	
Erdj4	Forward	5′-CCCCAGTGTCAAACTGTACCAG-3′/61 °C	117
	Reverse	5′-AGCGTTTCCAATTTTCCATAAATT-3′/56 °C	
Xbp-1	Forward	5′-GAACCAGGAGTTAAGAACACG-3′/57 °C	179
	Reverse	5′-AGGCAACAGTGTCAGAGTCC-3′/60 °C	
Atf4	Forward	5′-GCAAGGAGGATGCCTTTTC-3′/57 °C	100
	Reverse	5′-GTTTCCAGGTCATCCATTCG-3′/57 °C	
GAPDH	Forward	5′-CTTCAACAGCAACTCCCACTCTTCC-3′/64 °C	171
	Reverse	5′-GGGTGGTCCAGGGTTTCTTACTCCTT-3′/66 °C	

**Table 3 animals-15-01384-t003:** The composition of fatty acid in curd oil extracted from BSFL.

Fatty Acid	Common Name	Content (%)
C8:0	Caprylic acid	0.0
C10:0	Capric acid	2.1
C12:0	Lauric acid	51.1
C14:0	Myristic acid	6.4
C16:0	Palmitic acid	12.1
C16:1	Palmitoleic acid	1.9
C18:0	Stearic acid	12.8
C18:1	Oleic acid	11.1
C18:2	Linoleic acid	1.4
C18:3	Linolenic acid	0.0
C20:0	Arachidic acid	0.1
C22:0	Behenic acid	1.0
Saturated		85.6
Unsaturated		14.4

## Data Availability

The original contributions presented in this study are included in the article/Appendix A. Further inquiries can be directed to the corresponding author(s).

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
