# Peer review of "Medium-Chain Fatty Acids Extracted from Black Soldier Fly (Hermetia illucens) Larvae Prevents High-Fat Diet-Induced Obesity In Vivo in C57BL/6J Mice"

_animals, 2025, doi:10.3390/ani15101384_

Round 1
Reviewer 1 Report
Comments and Suggestions for Authors
In this article authors have used MCT derived from BSFL in micr and reported their anti obesity effects in mice. Overall the results are encouraging but the manuscript should be improved in light of the comments.
Major comments:
- Overall, the writing needs to tone down where there are specific claims
- Many missing citations, especially in the introduction and discussion, need to be added.
- The introduction needs to be improved in light of recent literature, and a clear gap and its solution need to be incorporated.
- Discussion needs to be improved
Minor comments:
Line 9-10: Summary; since the anti-obesity is done in mice, need to tone down the conclusion on human. Use term 'may" and "after further investigation"
Line 35: give full NCD at first instance and then use abbreviations. do the same for others as well
Line 38: Obesity is a key cause of cancer. Need to correct this. Similarly, in many instances in the manuscript, disease risk factors are taken as etiologies.
Line 55-56: need citation. there also are many instances where citation are unnecessarily combined at the end of paragraphs or multiple sentences. Correct citations are very important.
Line 58:
Moreover, decreased insulin resistance and adiposity due to MCT-enhanced energy ex- penditure has been reported in animal and clinical experiments [11-14]. After this sentence link if there are other studies which report the effects of MCT on abesity i.e https://doi.org/10.1016/j.jnutbio.2025.109914. There is a lack of gap analysis on the topic in this portion. Cite relevant studies and identify what specifically was not know which is targetted here in this manuscript.
Line 61-71: Many citations are missing.
Line 76: Which effective microbe or a commercial formulation. need to be specific.
Line 80-90: no citation of the method used for extraction. Similarly recheck citations for other methods.
Line 157: Table 2; add Tm and citations to primers or the primers were designed in this study. amplifon size?
Line 331-356: Discussion part, no citation.
Author Response
Thank you for your helpful and insightful comments. Now, we reply for your comments as below.
Major comments:
- Overall, the writing needs to tone down where there are specific claims
- Many missing citations, especially in the introduction and discussion, need to be added.
- The introduction needs to be improved in light of recent literature, and a clear gap and its solution need to be incorporated.
- Discussion needs to be improved
Minor comments:
Line 9-10: Summary; since the anti-obesity is done in mice, need to tone down the conclusion on human. Use term 'may" and "after further investigation"
: Thank you for your kind and insightful comment. We have toned it down and corrected it.
Line 35: give full NCD at first instance and then use abbreviations. do the same for others as well
: Thank you for your kind and insightful comments. We have corrected it to its full name.
Line 38: Obesity is a key cause of cancer. Need to correct this. Similarly, in many instances in the manuscript, disease risk factors are taken as etiologies.
: Thank you for your kind and insightful comment. We have changed to sentence to “major risk factor for various disorders, including certain cancer”
Line 55-56: need citation. there also are many instances where citations are unnecessarily combined at the end of paragraphs or multiple sentences. Correct citations are very important.
: Thank you for your kind and insightful comment. We have carefully revised the paragraphs to ensure that citations are placed appropriately within the text, corresponding directly to the specific statements.
Line 58:
Moreover, decreased insulin resistance and adiposity due to MCT-enhanced energy ex- penditure has been reported in animal and clinical experiments [11-14]. After this sentence link if there are other studies which report the effects of MCT on obesity i.e https://doi.org/10.1016/j.jnutbio.2025.109914. There is a lack of gap analysis on the topic in this portion. Cite relevant studies and identify what specifically was not know which is targetted here in this manuscript.
: Thank you for your kind and insight comment. We have revised and expanded the relevant section to strengthen the gap analysis. We have clearly identified the specific knowledge gaps addressed in this study and cited additional relevant studies to better contextualize our research. Furthermore, we have incorporated the suggested reference.
Line 61-71: Many citations are missing.
: Thank you for your kind and insight comment. We have cited the missing references.
Line 76: Which effective microbe or a commercial formulation. need to be specific.
: Thank you for your kind and insight comment. We have clarified the description in the manuscript.
Line 80-90: no citation of the method used for extraction. Similarly recheck citations for other methods.
: Thank you for your kind and insightful comment. We have added the citation of the method.
Line 157: Table 2; add Tm and citations to primers or the primers were designed in this study. amplifon size?
: Thank you for your kind and insightful comment. We have inserted Tm and amplicon information. We have also cited the reference to primer
Line 331-356: Discussion part, no citation.
: Thank you for your kind and insightful comment. We have rewritten the Discussion section to improve and added the citations.
Reviewer 2 Report
Comments and Suggestions for Authors
Line 1-4: Title seems ok, however, I would suggest to add "in vivo" for methodological clarity e.g., "prevent high-fat diet-induced obesity in vivo in C57BL/6J mice".
Line 16: please replace parameter with parameters in sentence "……improving blood lipid parameter…………"
Line 17-18: please change "preventing obesity" to "prevention of obesity" in sentence ‘………………………………….demonstrate the potential of BSFL-MCTs as feed and food ingredients for the preventing obesity and related metabolic disorders in both animals and humans’
Line 23-24: are these your results ‘Lauric acid, the predominant fatty acid in BSFL crude lipids, constituted >50% of the total lipids’ if yes, please mention. If not, please remove from here
Please also provide details of experimental design, treatments for example number of mice, control diet comparison with experimental diet
Line 24-26: please provide p values when presenting results ‘Mice supplemented with BSFL-MCTs exhibited significantly lower weight gain and food efficiency 25 ratios (FER) than HFD-fed mice, despite similar food intake’
Line 26: please don’t start line with abbreviations ‘BSFL-MCT supplementation…’ use full term
Line 26-36: please provide p values where needed ‘BSFL-MCT supplementation also attenuated HFD-induced increases in triglycerides, total cholesterol, and low-density lipoprotein cholesterol levels while improving cardiac risk indices. Furthermore, BSFL- MCTs reduced serum glucose and leptin levels, mitigated hypothalamic endoplasmic reticulum (ER) stress marker expression, and decreased serum alanine transaminase levels, indicating protective effects against obesity-related metabolic dysregulation. These findings suggest that BSFL-MCTs enhance energy expenditure and thermogenesis, thereby contributing to effective weight management and obesity prevention. As a sustainable and functional lipid source, BSFL-MCTs hold promise as a feed additive for animals and as a dietary ingredient for preventing pet obesity, offering an innovative approach to combat obesity and its associated health risks.’
Line 29-30: reticulum (ER) stress the abbreviation ER is not being used in abstract again please remove it
Line 41-42: don’t give any statement without reference. Please provide citation at the end of sentence ‘Obesity is a chronic disease caused by an increased intake of fatty and high-carbohydrate diets and decreased exercise.’
Line 49: add citation ‘Obesity-related complications pose a risk to animal health’ that will be ‘ Wenhao Xing and Shijie Li. Fat Metabolism-related lncRNA and Target Regulation and Application Studies in Chickens’Pak Vet J, 2023, 43(3): 579-584’
Line 45: replace "key cause of cancer" with "a risk factor for cancer" that is more precise
Line 52-57: blind statements without references. Please provide citaitons ‘Obesity can be treated using various approaches, depending on its causes, severity, and associated comorbidities. However, prevention through lifestyle modifications, including dietary control and exercise, is considered the best strategy. In most cases, dietary interventions for preventing obesity focus on calorie restriction’
Line 60-62: please add citations ‘Medium-chain triglycerides (MCTs), which are triglycerides found in various sources, including animal milk fat and plant oils such as olive, palm, and coconut oils, are a fat source used in the food and beverage industry’ Rasheed et al., Prophylactic Effects of Methylene Blue, Coconut and Olive Oils Supplements on Hemato-Biochemical and Histo-pathological Parameters against p-Phenylenediamine Toxicity in Male Albino Rats.Pak Vet J, 2024, 44(3): 840-846 and Ndiaye, et al.,. 2024. Influence of the extraction process on the chemical composition and oxidation state of baobab (Adansonia digitata L.) seed oil. Journal of Global Innovations in Agricultural Sciences 12(1)::45-52.
Line 63: please specify species "enhance hepatic thermogenesis" e.g., "in rodents/humans"
Line 68-69: please cite ‘Increasingly common climate anomalies have caused dramatic decreases in plant oil 68 production, leading to significant global price volatility.’ That could be ‘Dossa, L.I.KE-T., M.K. Bashir, S. Hassan and K. Mushtaq. 2023. Impact of climate change on agricultural production in Burkina Faso, West Africa. Journal of Global Innovations in Agricultural Sciences 11(3):319-332’ and ‘Kioko, T.M., S. Ndirangu and W. Nyarindo. 2024. Evaluating effect of climate smart agricultural practices adoption on productivity of drought-tolerant pulses: Insights from dryland areas of Makueni County, Kenya. Journal of Global Innovations in Agricultural Sciences 12:799-809.
"environmental cleanup insects" Clarify terme.g., "waste-reducing insects"
Materials & Methods
Section 2.2 in Extraction section please specify pressure "pressed at 120℃" e.g., "at 120℃ under X MPa"). And in "vacuum distillation" Define vacuum level (e.g., "under 5 mmHg").
In discussion section please link lauric acid to MCTs earlier (e.g., "Lauric acid (C12:0), the predominant MCT in BSFL..."). and also Address muscle mass vs. fat loss explicitly (e.g., "Reduced weight gain likely reflects fat loss, not muscle catabolism, as AST/creatinine were stable."). I would also suggest to cite a pathway (e.g., "MCTs may suppress ER stress via [X] signaling") in Leptin/ER stress.
At the end I would request authors to add a schematic and summarize mechanisms (e.g., BSFL-MCTs → thermogenesis that reduce obesity). Please also report effect sizes (e.g., Cohen’s d) where possible. How could you justify your results and recommendation while no female mice in the experiment? At last, how could you justify the result because the duration of study is short HFD duration (12 weeks vs. 14+ in prior studies)?
Author Response
Thank you for your helpful and insightful comments. Now, we reply for your comments as below.
Line 1-4: Title seems ok, however, I would suggest to add "in vivo" for methodological clarity e.g., "prevent high-fat diet-induced obesity in vivo in C57BL/6J mice".
: Thank you for your kind and insightful comment. In response to your comment, we have inserted in vivo in title.
Line 16: please replace parameter with parameters in sentence "……improving blood lipid parameter…………"
: Thank you for your kind and insightful comment. We have corrected “parameter” to “parameters”.
Line 17-18: please change "preventing obesity" to "prevention of obesity" in sentence ‘………………………………….demonstrate the potential of BSFL-MCTs as feed and food ingredients for the preventing obesity and related metabolic disorders in both animals and humans’
: Thank you for your kind and insightful comment. We have corrected “preventing obesity” to “prevention of obesity”.
Line 23-24: are these your results ‘Lauric acid, the predominant fatty acid in BSFL crude lipids, constituted >50% of the total lipids’ if yes, please mention. If not, please remove from here
: Thank you for your kind and insightful comment. The statement regarding lauric acid (>50% of total lipid) is based on our own results presented in Table 3. We have mentioned the information in abstract.
Please also provide details of experimental design, treatments for example number of mice, control diet comparison with experimental diet
: Thank you for your kind and insightful comment. We have provided the experimental design in section 2.3. Animal experiments and 2.4 animal diet. We did not specify the information in the abstract due to word count limitations.
Line 24-26: please provide p values when presenting results ‘Mice supplemented with BSFL-MCTs exhibited significantly lower weight gain and food efficiency 25 ratios (FER) than HFD-fed mice, despite similar food intake’
: Thank you for your kind and insightful comment. We have provided the p values in caption of Figure 1.
Line 26: please don’t start line with abbreviations ‘BSFL-MCT supplementation…’ use full term
: Thank you for your kind and insightful comment. We have changed the abbreviation as full term.
Line 26-36: please provide p values where needed ‘BSFL-MCT supplementation also attenuated HFD-induced increases in triglycerides, total cholesterol, and low-density lipoprotein cholesterol levels while improving cardiac risk indices. Furthermore, BSFL- MCTs reduced serum glucose and leptin levels, mitigated hypothalamic endoplasmic reticulum (ER) stress marker expression, and decreased serum alanine transaminase levels, indicating protective effects against obesity-related metabolic dysregulation. These findings suggest that BSFL-MCTs enhance energy expenditure and thermogenesis, thereby contributing to effective weight management and obesity prevention. As a sustainable and functional lipid source, BSFL-MCTs hold promise as a feed additive for animals and as a dietary ingredient for preventing pet obesity, offering an innovative approach to combat obesity and its associated health risks.’
: Thank you for your kind and insightful comment. We have provided the p values in Figure captions. However, we did not specify the information in the abstract due to word count limitations.
Line 29-30: reticulum (ER) stress the abbreviation ER is not being used in abstract again please remove it
: Thank you for your kind and insightful comment. We have removed it.
Line 41-42: don’t give any statement without reference. Please provide citation at the end of sentence ‘Obesity is a chronic disease caused by an increased intake of fatty and high-carbohydrate diets and decreased exercise.’
: Thank you for your kind and insightful comment. We have added the reference.
Line 49: add citation ‘Obesity-related complications pose a risk to animal health’ that will be ‘ Wenhao Xing and Shijie Li. Fat Metabolism-related lncRNA and Target Regulation and Application Studies in Chickens’Pak Vet J, 2023, 43(3): 579-584’
: Thank you for your kind and insightful comment. We have inserted the reference.
Line 45: replace "key cause of cancer" with "a risk factor for cancer" that is more precise
: Thank you for your kind and insightful comment. We have corrected the sentence and changed “key cause of” to “a risk factor”.
Line 52-57: blind statements without references. Please provide citaitons ‘Obesity can be treated using various approaches, depending on its causes, severity, and associated comorbidities. However, prevention through lifestyle modifications, including dietary control and exercise, is considered the best strategy. In most cases, dietary interventions for preventing obesity focus on calorie restriction’
: Thank you for your kind and insightful comment. We have inserted the missing references.
Line 60-62: please add citations ‘Medium-chain triglycerides (MCTs), which are triglycerides found in various sources, including animal milk fat and plant oils such as olive, palm, and coconut oils, are a fat source used in the food and beverage industry’ Rasheed et al., Prophylactic Effects of Methylene Blue, Coconut and Olive Oils Supplements on Hemato-Biochemical and Histo-pathological Parameters against p-Phenylenediamine Toxicity in Male Albino Rats.Pak Vet J, 2024, 44(3): 840-846 and Ndiaye, et al.,. 2024. Influence of the extraction process on the chemical composition and oxidation state of baobab (Adansonia digitata L.) seed oil. Journal of Global Innovations in Agricultural Sciences 12(1)::45-52.
: Thank you for your kind and insightful comment. We have inserted the references.
Line 63: please specify species "enhance hepatic thermogenesis" e.g., "in rodents/humans"
: Thank you for your kind and insightful comment. We have specified species in the sentence, e. g., in mice.
Line 68-69: please cite ‘Increasingly common climate anomalies have caused dramatic decreases in plant oil 68 production, leading to significant global price volatility.’ That could be ‘Dossa, L.I.KE-T., M.K. Bashir, S. Hassan and K. Mushtaq. 2023. Impact of climate change on agricultural production in Burkina Faso, West Africa. Journal of Global Innovations in Agricultural Sciences 11(3):319-332’ and ‘Kioko, T.M., S. Ndirangu and W. Nyarindo. 2024. Evaluating effect of climate smart agricultural practices adoption on productivity of drought-tolerant pulses: Insights from dryland areas of Makueni County, Kenya. Journal of Global Innovations in Agricultural Sciences 12:799-809.
: Thank you for your kind and insightful comments. We have inserted the references.
"environmental cleanup insects" Clarify terme.g., "waste-reducing insects"
: Thank you for your kind and insightful comment. We have changed “environmental cleanup insects” to “waster-reducing insects”.
Materials & Methods
Section 2.2 in Extraction section please specify pressure "pressed at 120℃" e.g., "at 120℃ under X MPa"). And in "vacuum distillation" Define vacuum level (e.g., "under 5 mmHg").
In discussion section please link lauric acid to MCTs earlier (e.g., "Lauric acid (C12:0), the predominant MCT in BSFL..."). and also Address muscle mass vs. fat loss explicitly (e.g., "Reduced weight gain likely reflects fat loss, not muscle catabolism, as AST/creatinine were stable."). I would also suggest to cite a pathway (e.g., "MCTs may suppress ER stress via [X] signaling") in Leptin/ER stress.
: Thank you for your kind and insightful comments. We have revised the Discussion section to explicitly link lauric acid to MCFAs earlier in the paragraph. We have also addressed the relationship between weight gain and fat vs. muscle loss by stating that stable AST and creatinine levels suggest that reduced weight gain likely reflect fat mass reduction rather than muscle catabolism. Additionally, we have cited a potential pathway (PERK-eIF2α-ATF4 axis) by which MCTs, including lauric acid, may suppress ER stress. Furthermore, we have added an explanation on how ER stress may impair leptin signaling through the SOCS3-mediated inhibition of the JAK2-STAT3 pathway, as you suggested.
At the end I would request authors to add a schematic and summarize mechanisms (e.g., BSFL-MCTs → thermogenesis that reduce obesity). Please also report effect sizes (e.g., Cohen’s d) where possible. How could you justify your results and recommendation while no female mice in the experiment? At last, how could you justify the result because the duration of study is short HFD duration (12 weeks vs. 14+ in prior studies)?
: Thank you for your kind and insightful comments. We have revised each of your comments as follows:
- A schematic and summarize mechanisms
: As suggested, we have added a schematic diagram summarizing the proposed mechanisms. The schematic has been included as Figure 6.
- Effect sizes (e.g., Cohen’s d)
: We have calculated effect sizes (Cohen’s d) for outcomes such as body weight gain, FER, lipid profiles, serum glucose and leptin levels. The calculated effect sizes have been the Discussion section.
- Justification of results for use of only male mice
: We used only male C57BL/6J mice to minimize variability associated with hormonal fluctuation in females. This approach is consistent with other metabolic studies seeking to establish baseline efficacy. However, we agree that inclusion of female mice is critical for generalization, and we have m
- Justification for HFD duration
: We are aware that previous studies have often employed HFD feeding durations exceeding 14 weeks to induce more pronounced metabolic dysfunction and organ damage. However, in this investigation, a 12-week HFD regimen was sufficient to induce significant physiological and biochemical alterations, including marked increases in body weight, dyslipidemia, and ER stress-related markers. These outcomes are well-established metabolic stress associated with obesity. Moreover, the primary objective of this investigation was not to induce overt organ toxicity but to evaluate the preventive effects of BSFL-MCFAs on HFD-induced obesity and early-stage metabolic disturbance. In this context, the observed effects of MCFAs demonstrate their potential anti-obesity and metabolic regulatory functions within the employed experimental duration. Therefore, although a loner HFD exposure might yield additional pathological manifestations, we believe that the current 12-week model sufficiently captured the key metabolic alterations necessary to assess the efficacy of BSFL-MCFAs.
Reviewer 3 Report
Comments and Suggestions for Authors
General comments
The manuscript titled “Medium-chain triglycerides extracted from black soldier fly (Hermetia illucens) larvae prevent high-fat diet-induced obesity in C57BL/6J mice,” proposed for publication in Animals, presents a novel and timely investigation into the anti-obesity potential of BSFL-derived medium-chain triglycerides (BSFL-MCTs) in a murine model. The research addresses relevant themes of sustainable nutrition and metabolic health and contributes valuable insights into functional feed additives. However, a few revisions and clarifications are necessary before the manuscript is suitable for publication.
Below are specific comments:
Introduction
- The introduction is comprehensive and provides a strong rationale for the study. The authors successfully frame the relevance of MCTs in obesity prevention and the potential of insect-derived oils as sustainable alternatives.
- It would strengthen the argument to include a more explicit comparison between BSFL-MCTs and more conventional MCT sources (e.g., coconut or palm oils) in terms of composition and availability.
- The transition from discussing human and animal obesity to proposing BSFL-MCTs as a nutritional intervention could be better justified, particularly for general readers unfamiliar with insect-based feed research.
- I suggest adding a short paragraph with previous studies on the use of BSFL oil , and the association with MCT. Present the effect of fats extracted from black soldier fly larvae.
Materials & Methods:
- The experimental design is generally appropriate, and the inclusion of both physiological and molecular outcomes is commendable.
- Add a reference for the sample preparation for fatty acids analysis.
- Briefly present the method for fatty acid analysis
- Mention the specific analytical details for a GC method (e.g., column type and dimensions, oven program, etc). Mention the equipment used and the producer (city, country).
- Add a reference for the enzymatic analysis.
- Briefly present the enzymatic determination.
- Present the Friedewald formula and add a reference for it.
- I suggest adding the proximate composition of the ND and HFD diets, not only the ingredients.
- A major concern is the absence of detailed analytical methods employed. The manuscript fails to specify the analytical procedures, equipment models, or protocols used. This omission is unacceptable, as it significantly undermines the reproducibility of the study. For the study to meet the standards of scientific rigor, full details must be provided .Without this information, other researchers cannot replicate or validate the findings.
- Clarification is needed regarding the calorie equivalence between the HFD and HFD_MCT diets (Table 1). While MCTs replaced lard, it is unclear whether total caloric content and macronutrient profiles were matched across diets, which is essential for interpreting weight changes.
- The animal housing conditions are adequately described; however, it would be useful to specify whether cage enrichment was provided, as this may influence stress and feeding behavior.
Discussion
- The results are clearly presented and supported by relevant figures and tables. The notably reduced weight gain and food efficiency ratio (FER) in the HFD_MCT group—despite comparable food intake, stand out as especially convincing and indicative of a true metabolic effect.
-The lipid profile improvements (TG, TC, LDL) in the HFD_MCT group are well documented. It may be helpful to clarify why HDL was elevated in both HFD and HFD_MCT groups and whether this is consistent with the literature on MCT supplementation.
- The inclusion of hypothalamic ER stress markers strengthens the physiological interpretation. However, the authors could elaborate on how BSFL-MCTs might mechanistically influence ER stress pathways, perhaps referencing relevant studies on lauric acid or MCTs more broadly.
- Some assertions, particularly regarding future application in pets and humans, are somewhat speculative. The authors should temper this language or clearly label it as hypothetical.
Author Response
Thank you for your helpful and insightful comments. Now, we reply for your comments as below.
The manuscript titled “Medium-chain triglycerides extracted from black soldier fly (Hermetia illucens) larvae prevent high-fat diet-induced obesity in C57BL/6J mice,” proposed for publication in Animals, presents a novel and timely investigation into the anti-obesity potential of BSFL-derived medium-chain triglycerides (BSFL-MCTs) in a murine model. The research addresses relevant themes of sustainable nutrition and metabolic health and contributes valuable insights into functional feed additives. However, a few revisions and clarifications are necessary before the manuscript is suitable for publication.
Below are specific comments:
Introduction
- The introduction is comprehensive and provides a strong rationale for the study. The authors successfully frame the relevance of MCTs in obesity prevention and the potential of insect-derived oils as sustainable alternatives.
- It would strengthen the argument to include a more explicit comparison between BSFL-MCTs and more conventional MCT sources (e.g., coconut or palm oils) in terms of composition and availability.
: Thank you for your kind and insightful comments. As suggested, we have added a direct comparison between BSFL-derived MCFAs and conventional sources such as coconut and palm oils to strengthen the argument.
- The transition from discussing human and animal obesity to proposing BSFL-MCTs as a nutritional intervention could be better justified, particularly for general readers unfamiliar with insect-based feed research.
- I suggest adding a short paragraph with previous studies on the use of BSFL oil, and the association with MCT. Present the effect of fats extracted from black soldier fly larvae.
: Thank you for your kind and insightful comments. We have inserted a short paragraph summarizing previous studies on BSFL oil and its association with MCFAs, including their reported metabolic effects in animal models, to better support the rationale for our intervention.
Materials & Methods:
- The experimental design is generally appropriate, and the inclusion of both physiological and molecular outcomes is commendable.
- Add a reference for the sample preparation for fatty acids analysis.
: Thank you for your kind and insightful comments. We have added the reference for the sample preparation for fatty acid analysis.
- Briefly present the method for fatty acid analysis
: Thank you for your kind and insightful comments. We have presented the method for fatty acid analysis.
- Mention the specific analytical details for a GC method (e.g., column type and dimensions, oven program, etc). Mention the equipment used and the producer (city, country).
: Thank you for your kind and insightful comments. We have mentioned the specific analytical details for a GC method and presented the equipment used and the producer
- Add a reference for the enzymatic analysis.
: Thank you for your kind and insightful comments. We have inserted the references for the enzymatic analysis.
- Briefly present the enzymatic determination.
: Thank you for your kind and insightful comments. We have briefly presented the enzymatic determinations.
- Present the Friedewald formula and add a reference for it.
: Thank you for your kind and insightful comments. We have added the Friedewald formula and its reference.
- I suggest adding the proximate composition of the ND and HFD diets, not only the ingredients.
- A major concern is the absence of detailed analytical methods employed. The manuscript fails to specify the analytical procedures, equipment models, or protocols used. This omission is unacceptable, as it significantly undermines the reproducibility of the study. For the study to meet the standards of scientific rigor, full details must be provided .Without this information, other researchers cannot replicate or validate the findings.
: Thank you for your kind and insightful comments. We have added specific information on the analytical methods and equipment based on your comment.
- Clarification is needed regarding the calorie equivalence between the HFD and HFD_MCT diets (Table 1). While MCTs replaced lard, it is unclear whether total caloric content and macronutrient profiles were matched across diets, which is essential for interpreting weight changes.
: Thank you for your kind and insightful comments. We have added the proximate composition of the ND, HFD, and HFD_MCFA diets (supplementary) to clarify macronutrient content. We also clarified in the Methods section that HFD-MCFA was matched to HFD in both caloric value and macronutrient ration, with only the fat source (lard vs. MCFAs) differing.
- The animal housing conditions are adequately described; however, it would be useful to specify whether cage enrichment was provided, as this may influence stress and feeding behavior.
: Thank you for your kind and insightful comments. We have added a description specifying that environmental enrichment (e.g., paper rolls, hut) was provided in all cages to minimize stress and support natural behaviors.
Discussion
- The results are clearly presented and supported by relevant figures and tables. The notably reduced weight gain and food efficiency ratio (FER) in the HFD_MCT group—despite comparable food intake, stand out as especially convincing and indicative of a true metabolic effect.
-The lipid profile improvements (TG, TC, LDL) in the HFD_MCT group are well documented. It may be helpful to clarify why HDL was elevated in both HFD and HFD_MCT groups and whether this is consistent with the literature on MCT supplementation.
- The inclusion of hypothalamic ER stress markers strengthens the physiological interpretation. However, the authors could elaborate on how BSFL-MCTs might mechanistically influence ER stress pathways, perhaps referencing relevant studies on lauric acid or MCTs more broadly.
- Some assertions, particularly regarding future application in pets and humans, are somewhat speculative. The authors should temper this language or clearly label it as hypothetical.
: Thank you for your kind and insightful comments. We have clarified the potential reason for elevated HDL levels in both HFD and HFD_MCFA groups, citing relevant literature on MCFA-induced changed in lipid metabolism. Additionally, we have provided additional explanation regarding the potential mechanisms through which BSFL-MCFAs, particularly lauric acid, may alleviate hypothalamic ER stress, with supporting references. Furthermore, speculative statements regarding future applications in pets and humans have been revised to clearly indicate that they are hypothetical and require further validation.
Round 2
Reviewer 1 Report
Comments and Suggestions for Authors
Manuscript improved now. I still have few minor comments
1) remove word may from title
2) table 2. I wanted to clarify that if you designed the primers for studying the gene expression or you used previously reported primers. no need for accession nos. it can be explained in text in relevant section that primers were designed using ref. sequences retrieved from NCBI.
Author Response
1) remove word may from title
: Thank you for your kind and insightful comments. We have removed the word “may” in title.
2) table 2. I wanted to clarify that if you designed the primers for studying the gene expression or you used previously reported primers. no need for accession nos. it can be explained in text in relevant section that primers were designed using ref. sequences retrieved from NCBI.
: Thank you for your kind and insightful comments. We have removed accession No and specified the source of primer sequences with an appropriate reference.
Reviewer 2 Report
Comments and Suggestions for Authors
Thanks for revision
Author Response
Thank you for your decision.